# Platinum Metallization of Polyethylene Terephthalate by Supercritical Carbon Dioxide Catalyzation and the Tensile Fracture Strength

**DOI:** 10.3390/ma16062377

**Published:** 2023-03-16

**Authors:** Po-Wei Cheng, Tomoyuki Kurioka, Chun-Yi Chen, Tso-Fu Mark Chang, Wan-Ting Chiu, Hideki Hosoda, Kei Takase, Hiroshi Ishihata, Hiromichi Kurosu, Masato Sone

**Affiliations:** 1Institute of Innovative Research, Tokyo Institute of Technology, Yokohama 226-8503, Japan; 2Diagnostic Radiology, Tohoku University Graduate School of Medicine, Sendai 980-8575, Japan; 3Division of Periodontology and Endodontology, Department of Ecological Dentistry, Tohoku University Graduate School of Dentistry, Sendai 980-8575, Japan; 4Cooperative Major in Human Centered Engineering, Nara Women’s University, Kitauoya Nishimachi, Nara 630-8506, Japan

**Keywords:** PET, platinum, supercritical carbon dioxide, electrical resistance, adhesion test, fracture strength

## Abstract

Polyethylene terephthalate (PET) is known to be highly inert, and this makes it difficult to be metallized. In addition, Pt electroless plating is rarely reported in the metallization of polymers. In this study, the metallization of biocompatible Pt metal is realized by supercritical CO_2_ (sc-CO_2_)-assisted electroless plating. The catalyst precursor used in the sc-CO_2_ catalyzation step is an organometallic compound, palladium (II) acetylacetonate (Pd(acac)_2_). The electrical resistance is evaluated, and a tape adhesion test is utilized to demonstrate intactness of the Pt layer on the PET film. The electrical resistance of the Pt/PET with 60 min of the Pt deposition time remains at a low level of 1.09 Ω after the adhesion test, revealing positive effects of the sc-CO_2_ catalyzation step. A tensile test is conducted to evaluate the mechanical strength of the Pt/PET. In-situ electrical resistances of the specimen are monitored during the tensile test. The fracture strength is determined from the stress value when the short circuit occurred. The fracture strength is 33.9 MPa for a specimen with 30 min of the Pt deposition time. As the Pt deposition time increases to 45 min and 60 min, the fracture strengths reach 52.3 MPa and 65.9 MPa, respectively. The promoted fracture strength and the decent electrical conductivity demonstrate the advantages toward biomedical devices.

## 1. Introduction

Flexible functional materials have received a great amount of attention to meet demands in various electronic industries, such as flexible displays [1], foldable sensors [2], organic light-emitting diodes (OLED) [3] and flexible supercapacitors [4]. Electronic components with flexibility overcome limitations in conventional rigid electronic components to allow for the continuous progress of electronics. Flexible functional materials can be achieved by combing a flexible substrate with a material possessing a desired function. Polymeric materials such as polyimide [5], polyamide 6 [6], polypropylene [7] and silk [8] are ideal materials as the flexible substrate. Materials with a decent electrical conductivity are always required for electronic devices. However, polymeric materials are usually electrically nonconductive. Thus, the combination of flexible polymers and electrically conductive metals is a solution to achieve flexible and electrically conductive functional materials for flexible electronics. Nickel phosphorus [9], gold [10] and platinum [11,12,13] are examples of electrically conductive functional materials to combine with flexible polymers.

There are a number of processes to combine a flexible polymer and an electrically conductive metal, such as sputtering [14], photodeposition [15], chemical vapor deposition [16], chemical fluid deposition [17], cold-spray coating [18] and electroless plating [19,20]. Among the metallization methods, the electroless plating method is the most suitable method because of the ease in controlling properties of the deposited metal layer, the ability to deposit metal layers on complex surfaces and the low process cost. In addition, the electroless plating process has already been demonstrated in various applications, such as corrosion resistance enhancement [21], catalytic electrodes [22] and flexible photocatalysts [23].

The electroless plating process commonly consists of three steps: a pretreatment step, a catalyzation step and a metal deposition step. The purpose of the pretreatment step is to clean and roughen surfaces of the substrate to enhance the interaction between the substrate surface and the catalyst seeds deposited in the catalyzation step. The catalyzation step is conducted to decorate catalyst seeds on the electrically nonconductive substrate surfaces. Then, in the deposition step, deposition of metals would be initiated from the catalyst seeds and gradually cover the entire substrate surface. On the other hand, corrosive aqueous solutions are used in the pretreatment step and the catalyzation step in the conventional electroless plating process. The substrate structure is often damaged in these two steps. Moreover, due to the high surface tension and polarity of the aqueous solutions, the catalyst seeds are only deposited on the substrate surface, since polymers are often nonpolar. This is a major cause leading to the poor interaction between the metal layer and the polymer substrate, which results in low electrical conductivity and poor reliability. 

A catalyzation step involving usage of supercritical carbon dioxide (sc-CO_2_) as the solvent is proposed for the metallization of polymer materials [24,25]. CO_2_ reaches the supercritical state when the temperature and pressure are both above the critical point (31 °C and 7.4 MPa). Sc-CO_2_ is nonpolar and has properties of zero surface tension and high self-diffusivity [26]. These are all beneficial for the metallization of polymers by allowing for transfers of catalyst seeds into the nonpolar polymer structure [24], which can be understood as planting catalyst seeds inside the polymer structure; then, the metal root-like structure would be formed from the catalyst seed inside the polymer structure and connected to the metal layer on surfaces of the polymer. By planting catalyst seeds inside the polymer structure and anchoring the metal layer by the metal roots inside the polymer structure, interactions between the metal layer and the polymer substrate would be significantly enhanced.

Polyethylene terephthalate (PET) is a massively produced polymer and is widely used in daily life due to its high mechanical strength, high chemical stability and low cost. PET is a common substrate in flexible electronics [27,28]. Meanwhile, the high chemical stability and density, as well as the smooth surface, are challenges in the integration of PET with functional materials. In particular, the adhesion between the metal layer and the PET surface is an issue, even when the PET is covered with a layer of metal. 

In this study, the Pt metallization of PET films is conducted by sc-CO_2_-assisted electroless plating. The source of the catalyst is an organometallic compound, palladium (II) acetylacetonate (Pd(acac)_2_), for its sufficient solubility in sc-CO_2_. After the sc-CO_2_ catalyzation step, a layer of highly biocompatible Pt is deposited on the PET to demonstrate the applicability toward medical devices. Reliability of the Pt-metallized PET film is evaluated by an adhesion test. The stretchability of the Pt-metallized PET is performed by a tensile test. There is limited reporting on the Pt metallization of PET. To the best of our knowledge, this is the first study evaluating the tensile strength of Pt-metallized PET composite materials.

## 2. Experimental Section

### 2.1. Materials

PET films (Toray Industries, Inc.: Tokyo, Japan) were cut into dog-bone-shaped specimens, as illustrated in Figure 1. In the narrow section, the length was 18 mm, the width was 3 mm and the thickness was 0.1 mm. The metallization of the PET by a conventional catalyzation step (CONV) was conducted to be used as a comparison with the sc-CO_2_ catalyzation step. A commercial catalyzation solution (Okuno Chemical Industries Co., Ltd.: ICP Accera KCR; Tokyo, Japan) was used for the CONV. For the sc-CO_2_ catalyzation, CO_2_ with a purity of 99.99% was purchased from Nippon Tansan Gas Co., Ltd. Palladium(II) acetylacetonate (Pd(acac)_2_) and ε-caprolactam (99%) were purchased from Sigma-Aldrich. The Pt electroless plating solution was also a commercial solution (MATEX Japan Co., Ltd.: Non-Cyanide Electroless Platinum Solution. Shizuoka, Japan), and the solution contained potassium tetranitroplatinate (0.1~0.3%), sodium borohydride (0.05~0.3%) and others (0.5~0.8%). For the tape adhesion test, a 3M Scotch tape (3M: Scotch Magic Tape, 810-1-18D) was used. 

### 2.2. Conventional and Sc-CO_2_ Catalyzation

In the catalyzation for the CONV, the PET specimen was immersed in the catalyzation solution at 30.0 ± 1 °C for 30 min. For the sc-CO_2_ catalyzation step, the sc-CO_2_ equipment was provided by Japan Spectra Company. The reaction cell was made of stainless steel 316 lined with polyether ether ketone coatings on the inner wall, and the inner volume was 50.0 mL. More details of the sc-CO_2_ equipment are reported in a previous study [10]. A piece of the PET specimen was placed in the reaction cell with 50.0 mg of Pd(acac)_2_ and 30.0 mg of ε-caprolactam (99%). The sc-CO_2_ catalyzation was conducted for 2.0 h at 120.0 ± 1 °C and 15.0 MPa.

### 2.3. Metal Deposition

No additional treatment was conducted after the catalyzation step. The conventionally catalyzed and sc-CO_2_-catalyzed PET films were immersed into the Pt electroless plating solution for 30 min, 45 min and 60 min at 70.0 ± 1 °C.

### 2.4. Characterization

The surface morphology and cross section of the Pt-metallized PET (Pt/PET) were observed by an optical microscope (OM, KEYENCE: VHV-5000) and a scanning electron microscope (SEM, HITACHI: S-4300SE). The crystal structure and phase of the Pt/PET were identified by an X-ray diffractometer (XRD, Rigaku: Ultima IV). The thickness of the Pt layers was evaluated by observing the cross section by the SEM. The electrical resistance was examined by a four-point probe (Mitsubishi Chemical Analytech Co., Ltd.: MCP-T37; Kanagawa, Japan). The adhesion test was conducted by firmly sticking a piece of the 3M tape with a 1 kg load to the specimen surface [10]. After removing the load and peeling the 3M tape off the specimen surface in sequence, the electrical resistance was measured again. To ensure reliability of the result, the adhesion test was conducted three times using different samples prepared under the same conditions. The tensile test was conducted by a universal testing machine (Shimadzu Corp.: Autograph Instron–type AG-1kNI). The electrical resistance of the tensile specimen was monitored by an electrical resistance meter (HIOKI E.E. CORPORATION.: RM3544) during the tensile test. The fracture point of the specimen was determined by the point when the electrical resistance suddenly increased to an extremely high value as a result of short circuit. 

## 3. Results and Discussion

### 3.1. Pt Metallization of PET

Surface appearances of the CONV catalyzed and the sc-CO_2_-catalyzed PET specimens after 60 min of the Pt deposition time are shown in Figure 1b. The PET specimen by the CONV remained clear and transparent, showing only features of the as-received PET film. The CONV was suggested to be ineffective at catalyzing the PET specimen, and led to the unsuccessful metallization result. On the contrary, the sc-CO_2_-catalyzed PET specimen was fully covered with a layer of gray metal-like coatings after the Pt deposition step. The metal-like coatings were believed to be metallic Pt, and the metallized PET remained flexible after the Pt deposition step. 

XRD patterns of the as-received PET film and the Pt-metallized PET specimens prepared with the sc-CO_2_ catalyzation step are shown in Figure 2. Surfaces of the PET were entirely covered with a metal-like appearance when a Pt deposition time of longer than 30 min was used. Therefore, only specimens with Pt metallizations of 30 min, 45 min and 60 min were characterized. The peak at 2θ = 25° revealed the contribution from the PET [29]. Peaks at around 2θ = 39.9°, 46.4°, 67.7°, 82.5° and 85.9° were evidence of the face-centered cubic (FCC) crystal structure of Pt (JCPDS card No 87-0647). Relative intensities of the Pt peaks increased and the PET XRD peaks gradually weakened following an increase in the Pt deposition time, which revealed that the amount of Pt on the PET specimen increased with the Pt deposition time. The thickness of the Pt layer is described in a later section. The Pt (111) peak became the XRD peak with the highest relative intensity when the Pt deposition time reached 45 min. The relative intensities of the Pt peaks did not vary much as the Pt deposition prolonged, which suggested that the crystal structure did not change much as the Pt deposition time increased. 

Morphologies of the Pt/PET with 30 min, 45 min and 60 min of the Pt deposition time are shown in Figure 3. The surfaces were covered with small and relatively uniform-sized particles and large irregular-shaped particles. As the Pt deposition time was prolonged, there was no obvious change in the average sizes of the small uniform-sized and large irregular-shaped particles, and an increase in the density of the irregular shaped particles was observed. 

The thickness of the Pt layer was determined from the cross-sectional images. Examples of the cross-sectional images are shown in Figure 3d–f. The Pt layer thickness reached 0.20 μm after 30 min of the Pt deposition time, and it thickened to 0.32 μm and 0.41 μm at 45 min and 60 min, respectively, of the Pt deposition time was used. Figure 4a shows the relationship between the Pt deposition time and the Pt layer thickness. The Pt layer thickness increased almost linearly as the Pt deposition time increased. Regarding the Pt layer growth rate, a complete Pt coverage of the PET surface was not achieved when the Pt deposition time was shorter than 30 min. Hence, the Pt growth rate was determined using the 30 min, 45 min and 60 min specimens. The Pt growth rate was 0.0067 μm/min in the first 30 min of the Pt deposition. For the Pt deposition time of 30 min to 45 min, the Pt growth rate was 0.0080 μm/min, and the Pt growth rate decreased to 0.0060 μm/min in the next 15 min interval. A plot of the Pt deposition time versus the Pt growth rate is provided in Figure 4a. In electroless plating, an incubation time is usually needed before the metal deposition starts, and the growth rate would gradually decrease when the deposition time is long because of continuous consumption of the metal ions and reducing agents involved in the metal deposition [30]. Therefore, generally, the growth rate is expected to be low to almost zero during the period of the incubation time. Then, a constant growth rate is achieved before concentrations of the reactants in the plating solution drop to a certain level. In this study, the incubation time was suggested to be less than 30 min for the sc-CO_2_-catalyzed PET specimen. 

### 3.2. Electrical Resistance of Pt/PET

The electrical resistances of the Pt/PET were summarized in Figure 4b. The Pt/PET prepared with 30 min of the Pt deposition time before the adhesion test showed a decent electrical resistance at 0.95 Ω with a small standard deviation. The result confirmed that the PET specimen was fully covered by the Pt. As the Pt deposition time extended to 45 min and 60 min, the electrical resistances were reduced to 0.80 Ω and 0.54 Ω, respectively. The reduced electrical resistance was a result of an increase in the amount of Pt on the PET specimen, and this corresponded well to the trend of the relative XRD peak intensity observed from the XRD characterization. 

Some parts of the metallized Pt could be removed from the PET surface by the peeling motion of the tape in the tape adhesion test if the adhesion between the Pt and PET was weak. Hence, the tape adhesion test was used to evaluate the reliability of the Pt layer on the PET. After the tape adhesion test, the electrical resistance of the Pt/PET with 30 min of the Pt deposition time increased to 1.76 Ω, which was an 85.3% increase. This increase in the electrical resistance was large, but the electrical resistance was still low and applicable toward electronic devices. For the Pt/PET with 45 min and 60 min of the Pt deposition time, the electrical resistances increased to 1.50 Ω and 1.09 Ω, respectively, after the adhesion test. The increasing percentages were 87.5% and 101.9% for the Pt/PET with 45 min and 60 min of the Pt deposition time, respectively. Again, the Pt/PET with the longest Pt deposition time showed the lowest electrical resistance after the adhesion test, but the increasing percentage was the highest. 

The resistance to the peeling motion of the tape in the tape adhesion test is suggested to be related to the size of the tape, the weight applied in the tape adhesion test, the interactions between the Pt layer and the PET substrate surface, and the surface conditions of the Pt layer. The size of the tape and the weight were the same in evaluations of all specimens. The interaction between the Pt layer and the PET substrate is suggested to be mostly affected by the sc-CO_2_ catalyzation. Again, the same sc-CO_2_ catalyzation step was applied in the 30 min, 45 min and 60 min specimens. Therefore, the surface condition illustrated in Figure 3 is thought to be the main cause leading to the increasing percentage of the electrical resistance after the tape adhesion test. The large irregularly shaped particles are suggested to be more likely to be removed after the tape adhesion test, leading to an increase in the electrical resistance. Hence, the 60 min specimen showed the largest increasing percentage after the tape adhesion test. 

### 3.3. Fracture Strength

Engineering stress–strain (SS) curves generated from the tensile test and the in-situ electrical resistances as the tensile test proceeded are provided in Figure 5. A video was recorded during the tensile test and provided as Appendix A. In the three SS curves, the yield point is not clear, and the linearity in the elastic region is poor. The specimens were composite materials composed of an outer Pt layer and PET core in the cross section, and this resulted in an unclear yield point and poor linearity. Regarding the in-situ electrical resistances in the three specimens, all three of them increased almost linearly before the short circuit occurred. The short circuit was defined to be the condition when the in-situ electrical resistance suddenly reached an extremely high value during the tensile test.

For the mechanical strength, the unclear yield point and the poor linearity lead to difficulties in determination of the yield strength. The electrical resistance is a key property indicating performance of the Pt/PET in this study. Therefore, the stress value when the short circuit occurred between two ends of the dog-bone-shaped specimen was used and reported as the fracture strength in this study. The fracture strengths are provided in Table 1. The fracture strength was 33.9 MPa for the Pt/PET with 30 min of the Pt deposition time. As the Pt deposition time increased to 45 min and 60 min, the fracture strength reached 52.3 MPa and 65.9 MPa. 

The overall strength of a composite material is expected to increase when there is an increase in the amount of the constituent material with a higher strength. The tensile strength of Pt is much higher than that of PET. Hence, an increase in the amount of the Pt layer in the Pt/PET is expected to cause an increase in the fracture strength. As shown in Figure 4a, the Pt layer thickness increased almost linearly with the Pt deposition time. Therefore, a positive relationship between the fracture strength and the Pt deposition time was an expected result.

## 4. Conclusions

Pt metallization of PET film was successfully achieved by electroless plating with the supercritical CO_2_ catalyzation step. Sc-CO_2_ functioned as the solvent, and Pd(II) acetylacetonate was used as the Pd catalyst source in the catalyzation step. The thickness of the Pt layer increased linearly as the Pt deposition increased from 30 min to 60 min. The Pt layer thickness reached 0.20 µm after 30 min of the metal deposition time, and the Pt layer thickened to 0.41 µm as the deposition time extended to 60 min. For the as-metallized Pt/PET with 30 min of the Pt deposition time, the electrical resistance was 0.95 Ω, and the electrical resistance reduced to 0.54 Ω as the Pt deposition time prolonged to 60 min. The thickened Pt layer was the main course of the reduced electrical resistance. For the Pt/PET with 60 min of the Pt deposition, the electrical resistance merely changed to 1.09 Ω after the tape adhesion test, and the fracture strength was at 65.9 MPa. The low electrical resistances before and after the adhesion test and the high tensile strength in the Pt/PET reported in this study all revealed advantages of the sc-CO_2_ catalyzation step in the development of biocompatible and flexible electronics. 

## Figures and Tables

**Figure 1 materials-16-02377-f001:**
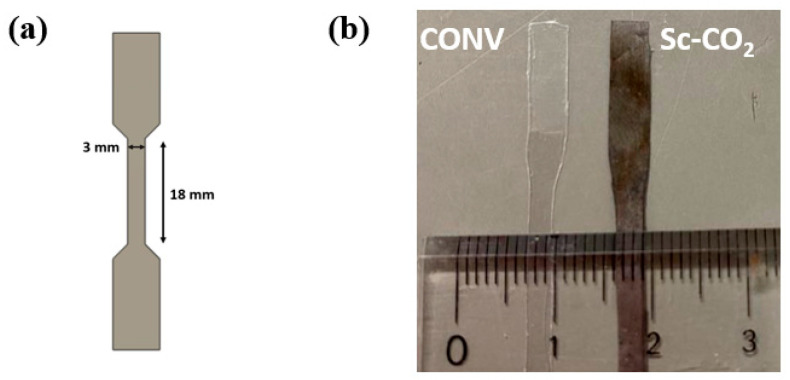
(**a**) Illustration of the dog-bone-shaped specimen. (**b**) Appearances of comparison of metalized PET catalyzed in conventional solution and sc-CO_2_.

**Figure 2 materials-16-02377-f002:**
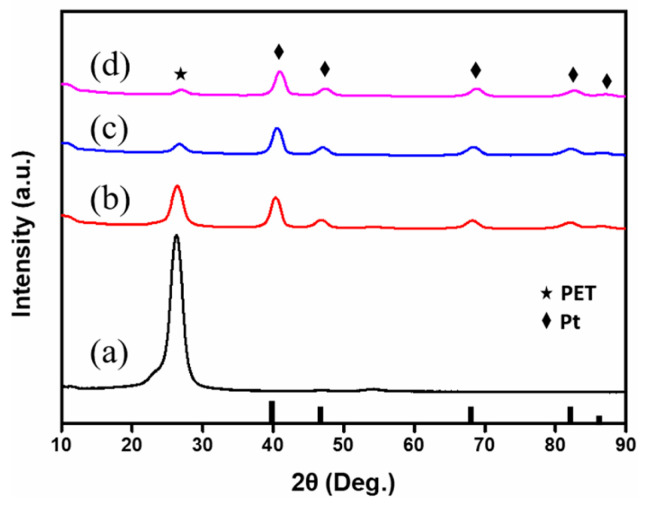
XRD patterns of the (**a**) PET film and the Pt/PET prepared with (**b**) 30 min, (**c**) 45 min and (**d**) 60 min of the Pt deposition time. (JCPDS card No 87-0647).

**Figure 3 materials-16-02377-f003:**
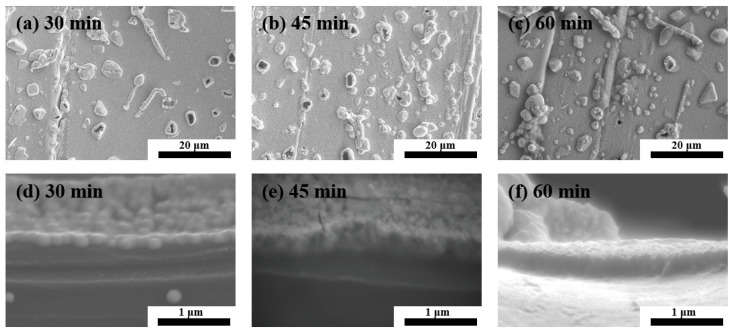
Surface morphology of Pt metallized PET treated by the sc-CO_2_ catalyzation followed by (**a**) 30 min, (**b**) 45 min and (**c**) 60 min of the Pt deposition time, and cross-sectional view of the specimen with (**d**) 30 min, (**e**) 45 min and (**f**) 60 min of the Pt deposition time.

**Figure 4 materials-16-02377-f004:**
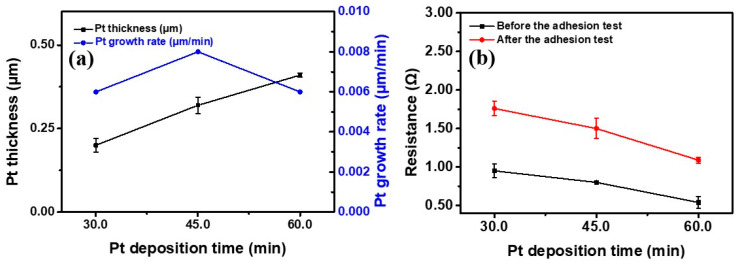
Plots of the (**a**) Pt deposition time versus the Pt layer thickness and the Pt growth rate, and the (**b**) Pt deposition time versus the electrical resistance.

**Figure 5 materials-16-02377-f005:**
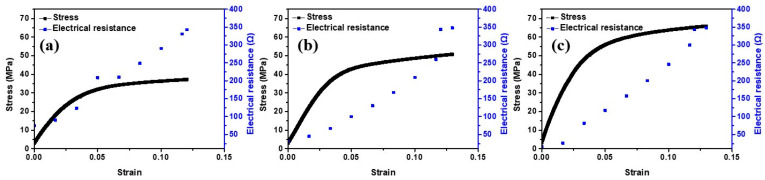
Engineering stress–strain curves and in-situ electrical resistances of Pt/PET prepared with (**a**) 30 min (**b**) 45 min (**c**) 60 min of the Pt deposition time.

**Table 1 materials-16-02377-t001:** Fracture strength of the Pt/PET.

Pt Deposition Time (min)	Pt Thickness (µm)	Fracture Stress (MPa)
30	0.20 ± 0.019	33.9 ± 2.84
45	0.32 ± 0.024	52.1 ± 3.42
60	0.41 ± 0.006	65.7 ± 8.96

## Data Availability

The data presented in this study are available on request from the corresponding author.

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
