# Peer review of "Platinum Metallization of Polyethylene Terephthalate by Supercritical Carbon Dioxide Catalyzation and the Tensile Fracture Strength"

_materials, 2023, doi:10.3390/ma16062377_

Round 1

Reviewer 1 Report

o   Materials-2252891

o   In the manuscript titled “Platinum Metallization of Polyethylene Terephthalate by Supercritical Carbon Dioxide Catalyzation and the Tensile Fracture Strength” authors have described the Metallization of polyethylene terephthalate (PET) with biocompatible Pt metal via supercritical CO2 (sc-CO2) assisted electroless plating. They evaluated the electrical resistance, and utilized a tape adhesion test to demonstrate intactness of the Pt layer on the PET film. The work is adequate and can be published in Catalyst after minor revisions.

Comments are:

o   Your abstract didn’t begin with a brief but precise statement of the problem or issue only it has a description of the research. in addition, you should add the significant findings, and the conclusions reached to your abstract.

o   Authors described that “Nickel phosphorus, gold and platinum are examples of the electrically conductive functional material to combine with the flexible polymer”. Then why Pt was chosen, even though it is costly material.

o   The introduction and results and discussion method should be enriched by citing some recent works.

o   Explain the novelty of the current work at the end of the introduction.

o   For what “CONV” stands?

o   Fig. 2 The XRD pattern must be improved.

o   The XRD pattern of the material should be supported by the previous work. No references are in XRD discussion?

o   Improve the English grammar and writing skills for this paper.

o   Add a comparative study table containing the similar previous studies.

Reviewer 2 Report

The manuscript entitled “Platinum Metallization of Polyethylene Terephthalate by Supercritical

Carbon Dioxide Catalyzation and the Tensile Fracture Strength” discusses about the metallization of PET substrate using biocompatible Pt. The works reports on the fundamental aspects, however, more clarity is required. The following comments may help to improvise the manuscript.

Abstract

1.      It’s highly recommended to discuss the problem statement in 1-2 first sentences of the abstract. When I read your abstract, I don’t know what’s the importance of metallization of PET.

Introduction

2.      Page 2, Line 79 – 84: Authors discussed about the challenges in fabricating metal on the surface of PET, but they did not address how to solve the issue, and authors must relate how the present work can address this issue.

3.      Page 2, Line 85 – 91: Brief discussion on why Pt is biocompatible and why Pt is selected as the metal of interest among the vast diversity choice of metals.

Methods

4.      Sufficient descriptions found in the methods.

Results and Discussion

5.      XRD – authors mentioned relative intensity of Pt peaks doesn’t change upon the increase of deposition time. However, an obvious Pt peak at 39.9 intensity seems to be enhanced as the deposition time increases. In addition, authors justification on the reduced intensity for PET seems to be qualitative. Unless, author justify their statement with appropriate qualitative measurement, for instance, ratio of PET peak/Pt peak.

6.      Optimum growth rate was recorded at 45 min, but why authors selected less than 30 mins as the optimum time for the growth rate? Justify.

7.      What is the importance and inter relation between tape adhesion test and resistance?

8.      Does the adhesion test followed any ASTM? Authors must relate with previous adhesion test studies like in this work: https://doi.org/10.1002/pc.22595. Kindly cite the relevant works.

9.      I don’t find any references or appropriate justification with previous works for the mechanical strength study. Improvise this section.

Reviewer 3 Report

Plagiarism was found to be 40 % after excluding references. It would be better if the editorial board checked for plagiarism issues before sending the manuscript out for review.

Metallization time increased, Pt layer thickness increased, fracture strength increased, and resistance decreased. The results seem to be very obvious. What does the study try to find? There are no optimization conditions and no clear goal for the study, both of which are missing.

What is the main scope of the present work? should be clearly described at the end of the introduction part.

Figure 3: The time denotation in the picture hides the images, need to be modified.

After the adhesion test, how the resistance is increased should be explained more clearly in the results and discussion section.

When the pt deposition time increased, Pt thickness increased with fracture stress. Why did the author not study at a lower time and beyond 60 min?

Reviewer 4 Report

The manuscript introduced Pt metallization of PET film achieved by electroless plating with the supercritical CO2 catalyzation step. The adhesion between the Pt layer and the PET and stretchability of the Pt metallized PET film were investigated in details. The comments are listed below.

1.    Introduction - “There is a number of processes to combine a flexible polymer and an electrically conductive metal……”. The photodeposition method can be used to prepare the composites. Some important references should be cited and explained for better understanding of photodeposition, such as Ceramics International, 2023, 49(2): 2262-2271; Applied Surface Science 579 (2022): 152171.

2.    Is there a size design requirement for the dog-bone shaped specimen?

3.    The authors should combine all XRD patterns into one image to better contrast the peak intensity of the samples at various deposition time.

4.    From the tensile strength of the Pt/PET, a positive relationship between the fracture strength and the Pt deposition time was found. The longer the deposition time, the higher the intensity. How to obtain optimal performance of Pt/PET?

5.    How about the strength of bonding interface between Pt and PET in this work?

6.    It is necessary to compare the performance of Pt/PET in this work and the reported work.

Round 2

Reviewer 3 Report

The paper can be accepted in its present form for publication.